# *Drosophila* as an Animal Model for Testing Plant-Based Immunomodulators

**DOI:** 10.3390/ijms232314801

**Published:** 2022-11-26

**Authors:** Andre Rizky Pratomo, Emil Salim, Aki Hori, Takayuki Kuraishi

**Affiliations:** 1Faculty of Pharmacy, Institute of Medical, Pharmaceutical and Health Sciences, Kanazawa University, Kanazawa 920-1192, Japan; 2Department of Pharmacology, Faculty of Pharmacy, Universitas Sumatera Utara, Medan 20155, Indonesia; 3AMED-PRIME, Japan Agency for Medical Research and Development, 1-7-1 Otemachi, Chiyoda-ku, Tokyo 100-0004, Japan; 4JST-FOREST, Japan Science and Technology Agency, Tokyo 102-0081, Japan

**Keywords:** *Drosophila melanogaster*, immunomodulator, plant extract, plant-derived compound

## Abstract

Allopathic medicines play a key role in the prevention and treatment of diseases. However, long-term consumption of these medicines may cause serious undesirable effects that harm human health. Plant-based medicines have emerged as alternatives to allopathic medicines because of their rare side effects. They contain several compounds that have the potential to improve health and treat diseases in humans, including their function as immunomodulators to treat immune-related diseases. Thus, the discovery of potent and safe immunomodulators from plants is gaining considerable research interest. Recently, *Drosophila* has gained prominence as a model organism in evaluating the efficacy of plant and plant-derived substances. *Drosophila melanogaster* “fruit fly” is a well-known, high-throughput model organism that has been used to study different biological aspects of development and diseases for more than 110 years. Most developmental and cell signaling pathways and 75% of human disease-related genes are conserved between humans and *Drosophila*. Using *Drosophila*, one can easily examine the pharmacological effects of plants/plant-derived components by employing a variety of tests in flies, such as survival, anti-inflammatory, antioxidant, and cell death tests. This review focused on *D. melanogaster*’s potential for identifying immunomodulatory features associated with plants/plant-derived components.

## 1. Introduction

The immune system plays an important role in defending the body against a variety of infectious and non-infectious substances that may induce severe illness or mortality [1]. The immune system is generally classified into innate and adaptive systems, each with distinct functions and purposes. Cells of the innate immune system utilize pattern recognition receptors (PRRs) to sense diverse pathogen-associated molecular patterns (PAMPs) and damage-associated molecular patterns (DAMPs) [2]. This response is very rapid and non-specific and includes responses such as phagocytosis, complement system, secretion of antimicrobial peptides to kill pathogens, and cytokine production. Additionally, adaptive immune responses comprise specialized T cells and B cells that fight various stimuli. The development of these lymphocyte-dependent adaptive immune responses is delayed; however, they are antigen-specific and result in long-term immunological memory [3].

In healthy organisms, the immune system maintains homeostasis. Unbalanced immune responses can cause various problems, including allergies, autoimmune diseases, immunosuppression, and acquired immunodeficiency syndrome [4]. Epidemiological research has indicated an upsurge in immunological illnesses. This has led to the development of a group of immunomodulators that can boost or weaken the immune response in diseases related to the immune system. Immunosuppressive drugs are used to suppress the immune response in many immunological-mediated disorders (i.e., in organ transplantation and autoimmune diseases), whereas immunostimulatory drugs are used to treat infections, immunodeficiency, and cancer [5]. Several monoclonal antibodies and chemically synthesized substances have been used as immunomodulators. However, owing to their severe adverse effects, there are significant restrictions on the general usage of these drugs [6]. Thus, immunomodulatory entities with better safety and efficacy are still required, and natural immunomodulators (i.e., plant/plant-derived compounds) are candidates as alternatives to chemical medicines in therapy regimens [6].

In the preclinical phase of drug discovery, commonly utilized animals include mice, rats, guinea pigs, rabbits, cats, and dogs [7]. These animal models facilitate the discovery of safe and effective treatments for certain diseases and/or their accompanying symptoms [8]. Nonetheless, rising concerns for animal welfare and rights have begun to restrict the use of these animals in preclinical research, imposing significant pressure on researchers to seek other model species for use in drug discovery studies [9]. Several model organisms have been developed as alternative in vivo platforms for studying the pathophysiology diseases [10]. *Drosophila*, a model invertebrate species, is becoming increasingly important in clinical drug development. Given the similarities between *Drosophila* and humans in the biochemical pathways that regulate several essential cellular processes, *Drosophila* can be used to assess the efficacy of novel drugs. Using *Drosophila*, new drugs can be tested considerably more rapidly than with mammalian models, and they can be utilized as an alternative to cell culture for the initial high-throughput screening procedure. Screening of a whole organism facilitates the identification of drugs with an improved safety profile for testing in costly mammalian models. Furthermore, it may be quite simple to manipulate the genetic background of *Drosophila* to imitate a pathological condition to investigate medication efficacy [11]. Utilizing *Drosophila* in chemical screening offers the added benefit of restricting the use of mammalian models, thereby reducing issues related to animal ethics [12]. For example, *Drosophila* Alzheimer’s Disease Model is considered promising for the screening and discovery of medicines [13]. This review focuses on the use of *Drosophila* to investigate the immunomodulatory effects of plant/plant-derived compounds.

## 2. *Drosophila* as a Model for the Study of the Effects of Plants/Plants Derived Compounds

*Drosophila melanogaster*, commonly known as the fruit fly, has been used as a model organism for biological studies for more than a century. It has become a valuable tool fundamental to understanding genes, chromosomes, and inheritance of genetic information [14,15]. Initially, fruit fly served as the primary model organism for classical genetics until the basic genetic rules and techniques developed during the first half of the 20th century were identified and utilized as a potent way to investigate biological phenomena [16]. In the past 50 years, fly genetics has been systematically and successfully applied to decipher the fundamental mechanisms underlying numerous fundamental biological processes, such as development [17], nervous system development, function, and behavior [14], and flies have become useful tools for studying human diseases, such as rare Mendelian diseases [18], neurodegenerative disorders [15] and cancer [19]. *Drosophila* has also been used to study innate immune responses, such as the role of Eater in phagocytosis [20], activation of the Toll pathway in response to fungi [21], involvement of the Immune deficiency (IMD) pathway against bacteria [22], and role of immune response in sterile inflammation [23]. Given the great degree of evolutionary conservation, this study has established an essential foundation for research on mammals, and the fly continues to serve this capacity [14,16].

The fly has several favorable features that make it a good model for studying medicinal plants. The genome of the fruit fly has been entirely sequenced and annotated. It has over 14,000 genes spread over four chromosomes, although only three of them contain the bulk of the genome [10,24]. Two-thirds of the known disease-causing genes in humans have been identified in the fly, and sequencing of both genomes has demonstrated remarkable gene and pathway conservation between flies and humans [24,25]. Therefore, it is a promising human illness model for studying the pharmacological effects of many medicinal plants. The genome information of *Drosophila* allows targeted tissue-specific overexpression and downregulation of disease-inducing genes that may be used to determine the medicinal/pharmacological effects of various plants/plant-derived components by examining their influence on disease progression and rescue [26]. The GAL4-UAS binary system is a sophisticated tool used to upregulate and downregulate a gene [27]. The GAL4/UAS system enables spatiotemporal control of the expression of modified genes that contain an Upstream Activation Sequence (UAS). Flies bearing the UAS transgene are mated with GAL4 lines, whose GAL4 transcriptional activator is controlled by a specific gene promoter (e.g., FB-GAL4 to drive expression in the fat body). Because GAL4 stimulates the transcription of the gene downstream of the UAS, any cell type expressing GAL4 will also express a transgene under UAS control [28]. Mutations of any gene in *D. melanogaster* can be easily generated within a month using the clustered regularly interspaced palindromic repeats/CRISPR-associated (CRISPR/Cas9) system, allowing the creation of a large number of mutant and transgenic fly lines. Injecting RNA into *Drosophila* embryos can elicit extremely efficient mutagenesis of the desired target genes in as many as 88% of injected flies [29]. Stocks of *Drosophila* cannot be frozen and all fly lines are kept viable. The fly lines can be obtained from other researchers or purchased from stock centers, such as the Bloomington *Drosophila* Stock Center at Indiana University, which maintains an extensive collection of mutants, RNAi, misexpression, and other stocks [30]. This facility manages over 79,000 *Drosophila* stocks and exports over 180,000 fly cultures in 2021. Additional stock centers have developed genome-wide RNAi collections, such as the Harvard Transgenic RNAi Project (TRiP) [31], Kyoto *Drosophila* Stock Centre [32], and Vienna *Drosophila* Research Center (VDRC) [33]. The availability of tools to precisely modify the expression of almost every gene has a profound impact on research capacity. With the high number of cell-type-specific drivers, as well as compound- or temperature-activated driver lines, precise spatiotemporal control of gene expression is easily accomplished in *Drosophila*. The genetic toolkit of *Drosophila* is more adaptable than that of any other multicellular organism and is continuously increasing, enabling more sophisticated manipulation of the *Drosophila* genome [15]. High-quality and comprehensive data on *Drosophila* genes and genomes curate published phenotypes, gene expression, genetic and physical interactions, and numerous other datasets pertaining to *Drosophila* genetics that can be accessed from FlyBase (www.flybase.org accessed on 2 July 2022) [34]. Additionally, the life cycle of a fly is short. Within 10–12 days at 25 °C, a single viable mating pair can generate hundreds of offsprings that are genetically identical to their parents. The traditional rodent models have much lower reproductive rates, with only a small number of offsprings born every 3–4 months [10]. Consequently, a large number of flies can be used to test the pharmacological effects of a given plant extract or an active compound. In addition, its relatively short lifespan of 90–120 days [26] facilitates the swift investigation of the impact of plant extract/active components on immunological function in age-related illnesses, such as neurodegenerative disorders. Moreover, *Drosophila* is a very small insect (approximately 3 mm in size), very easy to handle, and requires very little space in the laboratory; as a result, it is possible to grow a significant number of flies in both laboratories and stock centers [26]. Because of its high fecundity (a single female may lay between 30 and 50 eggs per day), it can conduct genetic and pharmacological testing on a large scale [26]. The phenotypic (eye, anomalies, etc.), developmental (life cycle, lifespan, fecundity larval/pupal/adult development), and behavioral (i.e., locomotor, climbing, phototaxis) effects of targeted compounds in flies (disease model) can be easily examined by feeding them a diet containing plant/plant-derived components [26].

*Drosophila*, like other animal models, has both benefits and disadvantages. *Drosophila* body size and organization differ from those of mammals, which presents certain challenges. For instance, owing to *Drosophila’s* lack of adaptive immunity, the model cannot be used to identify drugs involving adaptive immunity mechanism. However, despite the absence of the adaptive immunity in *Drosophila*, this model can be used to investigate the features of the innate immune system that would otherwise be masked by the adaptive immune response [35]. Blood arteries are also absent in *Drosophila*, which makes studying the effect of drugs on homeostasis challenging [9]. Despite these drawbacks, using invertebrate models, such as *Drosophila*, during the preliminary stages of drug screening has the potential to hasten the identification of positive hits at an affordable cost. Initially, the model of human disease in flies was generated either through mutation of the fly homolog of a human disease-related gene or by expression of the human form of the gene. This model can be used to screen small compounds that rescue the phenotype or are subjected to genetic screening to discover modifiers of the phenotype that represent new potential targets. After the initial screening, the positive hits can be validated using additional fly disease models. Next, drugs with efficacy in *D. melanogaster* models are subsequently validated in whole-animal disease models of mammals [10].

## 3. *Drosophila* Immune System

*Drosophila* is a powerful model to decipher the molecular mechanism of the host’s innate immune response to PAMPs and DAMPs and to screen the pharmacological effects of medicinal plant extracts and their derived compounds that may function in the immune response. This is because of the highly conserved innate immune system between flies and humans, such as the mechanism of pathogen recognition, immune pathways, and transcription factors [36,37,38,39,40,41]. *Drosophila* mount a complex innate immune response that includes humoral and cell-mediated mechanisms in response to microbial infections. The fat body, which is functionally analogous to the human liver, secretes antimicrobial peptides (AMPs) into the hemolymph in response to microbial PAMPs as part of the humoral response [37,42].

There are three main types of surveillance cells (hemocytes) in the *Drosophila* immune system: plasmatocytes, lamellocytes, and crystal cells. Approximately 95% of all circulating hemocytes are plasmatocytes, which are similar to the professional phagocytes found in mammals and are also involved in the generation of AMPs [37,43]. Plasmatocytes, like macrophages in mammals, become tissue-resident cells after terminal differentiation [44]. Crystal cells, which make up the remaining 5% of circulating hemocytes, release phenoloxidase cascade components necessary for melanization of invading pathogens and wound healing. The third cell type, lamellocytes, is the largest but least abundant cell type in circulation in healthy larvae. They play a role in the encapsulation of invading pathogens that are too large to be phagocytosed, such as wasp eggs. The crystal cells and lamellocytes found in *Drosophila* are not conserved in mammals [44].

*Drosophila* plasmatocytes release various cytokines in response to infection. Upon infection, plasmatocytes release unpaired 3 (upd3), which is homologous to human IL-6, leading to activation of the Janus kinase (JAK)-signal transducer and activator of transcription (STAT) cascade by binding to the JAK/STAT receptor Domeless (Dome) [45]. A subset of plasmatocytes also secrete transforming growth factor-β (TGF-β)-related cytokines, decapentaplegic (dpp) and dawdle (daw), in response to wounds and infections. In response to wound, dpp is rapidly activated and inhibits the generation of antimicrobial peptides, thereby acting as a major inhibitor of inflammation. Conversely, dawdle is triggered by Gram-positive bacterial infections and inhibit infection-induced melanization. Without infection, *daw* knockdown flies still showed a significant melanization response [46].

In *Drosophila*, the Toll and immune-deficiency (IMD) pathways are two distinct nuclear factor-kappa B (NF-κB) signaling pathways responsible for the generation of antimicrobial peptides in *Drosophila* in response to an infection [37]. The Toll pathway, the downstream pathway of the Toll receptor, mediates resistance to fungal and Gram-positive bacterial infections. In contrast to TLRs in vertebrates, the fly Toll receptor does not have a direct pathogen binding site. Rather, it is triggered by a protein found naturally in the body called Spätzle (Spz), which is a cysteine-knot protein with structural similarities to the nerve growth factor in humans. Binding of Spz to the Toll receptor leads to the recruitment of Myddosome, a protein complex consisting of dMyd88, Tube, and Pelle kinase, via the E3 ubiquitin ligase called Sherpa [47]. The formation of this complex results in the destabilization of the IκB protein, Cactus, via unidentified mechanisms. This, in turn, leads to the activation of the NF-κB proteins, Dorsal and Dif, responsible for the expression of antimicrobial genes such as *Drosomycin* [48]. The IMD pathway is required for the defense response of adult flies against Gram-negative bacterial infections. Gram-negative bacteria release diaminopimelic acid (DAP)-type peptidoglycans that activate peptidoglycan recognition protein LC (PGRP-LC) or PGRP-LE, transmembrane or intracellular pattern recognition receptors. This activation induces the recruitment of the adaptor protein IMD. Then, the IMD protein is cleaved by a caspase-like protein Dredd to form a complex containing the E3 ubiquitin ligase Diap2, which ultimately activates TAK1 and the IκB kinase (IKK) complex. The IKK complex phosphorylates and activates Relish, an NF-κB, to promote the expression of genes encoding antimicrobial proteins such as *Diptericin* [35,37].

The JAK/STAT pathway is an immune-related, evolutionarily conserved signaling pathway (Figure 1) [49,50]. In *Drosophila*, this pathway is essential for the defense against viral infection, midgut regeneration following bacterial infection, hematopoiesis, and lamellocyte differentiation in response to parasitic infestation [35]. In the Jak/STAT pathway, three ligands called unpaired (upd), upd2, and upd3 can bind to a single receptor, Dome, activating JAK, hopscotch (hop), and STAT transcriptional factor, STAT92E [35,37]. STAT92E is then phosphorylated, allowing its dimerization and nuclear translocation, where it is capable of binding to a palindromic response element and inducing target gene expression [51].

## 4. Examples of Studies to Test the Immunomodulatory Effect of Plants/Plant-Derived Compounds Using *Drosophila*

*Drosophila* has been used as an animal model to test plant extracts and their derivatives. Plants/plant active compounds can be fed orally by mixing them with normal food. *Drosophila* can be orally introduced to toxic compounds by mixing them with food and infectious agents (viruses, bacteria, and fungi) orally or systemically. The immunomodulatory effect of the tested plant extracts/active compounds can then be evaluated.

Li et al. (2013) studied the effects of *Aanthopanax senticosus* extract on intestinal immunity in wild-type adult *Drosophila* orally infected with bacteria and fed with toxic compounds. They found that *A. senticosus* extract increased the survival of the flies, decreased intestinal epithelial cell death, increased antimicrobial peptide gene expressions, and reduced melanotic mass formations [52]. Liu et al. (2016) investigated the effects of *Crocus sativus* L. extract on intestinal immunity in adult wild-type *Drosophila*. Their study revealed that the extract greatly extended the longevity and survival rate of adult flies. In addition, the extract may reduce epithelial cell death and reactive oxygen species (ROS) levels, thereby improving intestinal morphology [53]. Using adult wild-type *Drosophila*, Oboh et al. (2018) showed that *Gnetum africanum* extract inhibited Mn-induced elevation of NO and ROS levels [54]. To screen the protective effects of selected traditional plants on intestinal cells, Zhou et al. (2016) utilized adult wild-type *Drosophila* fed with toxic compounds. Their study indicated that *Codonopsis pilosula*, *Saussurea lappa*, *Imperata cylindrica*, and *Melia toosendan* water extracts increased fly survival, reduced epithelial cell death, and improved gut morphology. Additionally, *C. pilosula* extracts enhanced antimicrobial peptide expressions (*Dpt* and *Mtk*) following treatment with sodium dodecyl sulfate (SDS) [55]. Zhu et al. (2014) investigated the effects of *Rhodiola crenulata* extracts on gut immunity in adult wild-type *Drosophila*. The results indicated that *R. crenulata* improved the survival rates of *Drosophila* and increased the expression of antimicrobial peptide genes (*Def*, *Drs*, and *Dpt*) following ingestion of a pathogen or toxic compound. In addition, ROS levels and epithelial cell death were reduced, which are associated with improved intestinal morphology [56]. Ekowati et al. (2017) studied the protective effect of phytohemagglutinin (PHA) isolated from the family of *Phaseolus vulgaris* beans against viral infection, using adult wild-type *Drosophila* as a host. They found that the survival in *Drosophila* fed PHA-P, a mixture of L4, L3E1, and L2E2, was improved. Moreover, the expression levels of phagocytosis receptors in flies increased after feeding with PHA-L4 [57].

To study the effects of anthocyanins on tissue inflammation, Valenza et al. (2018) used a *Drosophila* model that mimics human adipose tissue macrophage (ATM) infiltration. Using the GAL4-UAS system, pupariation was prevented by creating larvae *P0206-Gal4; UAS-Ni* in which the reduction in the size of the prothoracic gland that produces ecdysone leads to reduced levels of ecdysone, resulting in animals that develop at a nearly normal rate and continue to feed for 3 weeks with increased body weight. These larvae acquired features of obese individuals, including elevated triglycerides (TAGs), glucose in the hemolymph, resistance of fat cells to insulin stimulation, and increased hemocytes in the fat body. Their study showed that anthocyanin reduced the infiltration of hemocytes into the fat body, reduces the production of ROS, and activated the JNK/SAPK p46 stress kinase [58]. Asfa et al. (2022) studied the immunosuppressive effects of bitter gourd (*Momordica charantia* L.) extract using PGRP-LB mutants. The lack of PGRP-LB has been demonstrated to stimulate overactivation of the NF-κB (IMD) pathway in flies. They found that the extract enhanced the survival and locomotion of PGRP-LB mutants in a concentration-dependent manner and reduced the expression of *Dpt* and *Dro*, the downstream genes in the IMD pathway [59].

To investigate the anti-melanogenic effects of arbutin and arbutin undecylenic acid ester, we had used wild-type *Drosophila* as the animal model [60]. Melanization is an important immune response in flies, which involves the synthesis of melanin to encapsulate pathogens [61]. However, excessive melanin formation has been linked to various skin disorders, including hyperpigmentation and skin cancer. We had pinched the 3rd instar larvae and had pricked the adult flies following compound treatment and had observed a blackening reaction at the wound site. These results indicated that arbutin undecylenic acid ester inhibits melanization. Utilizing a combination of *D. melanogaster* and *B. mori* as animal models, we successfully designed fast, cheap, and highly effective methods to screen tyrosinase inhibitors, agents that inhibit tyrosine enzymes that play an important role in the production of melanin [60]. We also utilized the *Drosophila* assay system to investigate the inhibitory effect of fungal decalin-containing diterpenoid pyrones (DDPs) on the Toll and IMD pathways, which are the frontlines of defense against microbial infection. *Drosophila* cells, embryonic macrophage-derived *Drosophila* DL1 cells, and larval blood cell-derived l(2)mbn cells were used to test the effect of DDPs on the Toll and IMD pathways, and compound 21 was found to inhibit the IMD pathway [62].

Padalko et al. (2020) investigated the effects of *Zingiber officinale* using the wild-type Oregon R strain *Drosophila* model. The flies were fed food containing dithiothreitol, a reducing agent that induces oxidative stress, and *Z. officinale* powder and were inspected daily to examine the lifespan of flies. In addition to increasing the life expectancy of flies, *Z. officinale* significantly reduced the negative effects of dithiothreitol and oxidative stress outcomes [63]. We had also briefly investigated the immunomodulatory effects of *Z. officinale* extract using *Drosophila* DL1 cells stably expressing the *Drosomycin*-firefly luciferase reporter. The cells were activated using larval extract containing Spätzle, a ligand that plays an important role in activating toll receptors in *Drosophila* [64]. *Z. officinale* extract was found to induce *Drosomycin* expression in cells, suggesting the activation of Toll receptor signaling in DL1 cells (unpublished observation).

The innate immune response in *Drosophila* is controlled by two primary signaling cascades, the Toll and IMD pathways, both of which activate members of the NF-κB family of transcription factors [37]. Chronic hyperactivation of NF-κB in immune cells is associated with neurodegenerative disorders. Thus, the identification of natural compounds and phytochemicals that can modulate NF-κB activity is of particular interest [65]. Previous studies have reported neuroprotective effects of plant-derived polyphenols in *Drosophila* Parkinson’s disease (PD) models [66,67]. In a *Drosophila* PD model, grape skin extracts were found to restore mitochondrial abnormalities, increase health, and extend lifespan [68]. It has been suggested that resveratrol, which is found in grape skin, acts as an anti-aging agent, and contributes to the health-promoting benefits of grape skin. Remarkably, the positive effects of resveratrol were observed in a 6-hydroxydopamine-induced PD rat model [69]. Additionally, in the α-syn *Drosophila* model of PD, the isoflavone genistein was found to extend lifespan and delay mobility impairments [70]. These studies demonstrate the potential use of *Drosophila* as an invertebrate model to screen for the neuroprotective effects of plant extracts and plant-derived compounds.

A summary and other examples of the immunomodulatory effects of plant/plant-derived compounds are summarized in Table 1.

## 5. Conclusions

Plant extracts have been traditionally used for their therapeutic or preventive effects in many debilitating disorders, including immune-related diseases. These extracts contain diverse and complex blends of bioactive compounds such as polyphenols, anthraquinones, and flavonoids, which may contribute to beneficial modulation of the immune system. *Drosophila* is an excellent model species that has been widely adopted for studies on most biological processes, including the immune system. Based on the availability of various powerful tools of both genetics and molecular biology, the fly model is a useful alternative model for studying the immunomodulatory effects of plants/plant-derived compounds.

## Figures and Tables

**Figure 1 ijms-23-14801-f001:**
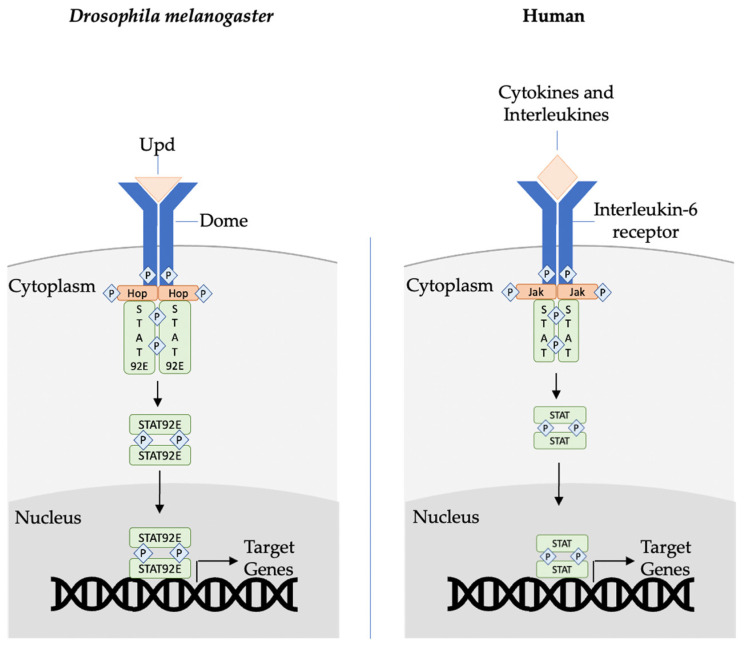
The Jak/STAT signaling core components of *Drosophila* and their human homologs. Upd (Interleukin or cytokine in humans) binds to its signaling receptor Dome (interleukin-6 receptor in humans), activating the associated Hop (Jak1, 2, 3, tyrosine kinase 2 in humans), and initiating a cascade of actions. Activated Jak phosphorylates other Jaks and the receptor, thereby forming a STAT protein-binding site. The phospho-STATs undergo dimerization and nuclear translocation. The STAT DNA-binding domain identifies promoter and enhancer regions of target genes, hence activating their transcription.

**Table 1 ijms-23-14801-t001:** Examples of immunomodulatory effects of plant/plant-derived compounds tested using the *Drosophila* model.

Name of Plants/Compounds	Parts of Plants	Solvent	Drosophila Stage	Age	Model	Experimental Systems	Dose	Effects	Refs.
*Aanthopanax senticosus*	Fruit	Water	Fly	3–5 d	Wild-type *w^1118^*	orally infected by *M. luteus* and *B. bassiana*, and orally fed with toxic compounds SDS, high osmolarity of NaCl, and CuSO_4_.	10% of medium	Increased survival rate Decreased death of intestinal epithelial cells Increased expression of AMPs genes (*Defensin*, *Drosomycin* and *Diptericin*) Decreased formation of melanotic masses	[52]
Anthocyanins	-	-	Larva	1st instar larvae (1 d)	*P0206-GAL4>UAS-Ni* with characteristics of obesity [triglycerides (TAGs), glucose circulating in the hemolymph, resistance of fat cells to stimulation with insulin, hemocytes in the Fat Body (Liver/adipose tissue)].	Fed with food containing anthocyanins	0.24 mg/mL	Decreased hemocytes infiltrating the fat cells Decreased ROS Decreased activation of the JNK/SAPK p46 stress kinase	[58]
*Crocus sativus* L.	Stigma	Water	Fly	3–5 d	Wild-type *w^1118^*	Fed with toxic compound (SDS, paraquat)	1% medium	Increased lifespan and survival rate Decreased epithelial cell death and ROS levels, resulting in improved intestinal morphology.	[53]
*Gnetum africanum*	Leave	Alcoholic-acetic acid	Fly	3–5 d	Wild-type Harwich strain	Fed with food containing Mn	2.5 mg/g medium	Increased survival rate and locomotor performance Decreased AChE activity, NO, and ROS levels	[54]
50 selected plants	Not mentioned	Water	Fly	4–5 d	Wild-type *w^1118^*	Fed with food containing SDS or DSS	1.25–10% medium	Some extracts: increased survival rate epithelial cell death ↓ improved gut morphology *C. pilosula* extracts: Increased expression of *Dpt* and *Mtk*)	[55]
*Momordica charantia* L.	Fruit	Ethanol	Fly	5–7 d	Oregon R (wildtype) and PGRP-LB^Δ^ (mutant line lacking PGRP-LB expression), the lack of PGRP-LB has been demonstrated to stimulate the overactivation of NF-κB (IMD) pathway in *D. melanogaster*	Fed with food containing extract	0.02, 0.2, and 2% medium	Increased survival rate and locomotor performance Decreased expression of *dpt* and *dro*, downstream genes in the *Drosophila* IMD (NF-κB) pathway	[59]
*Rhodiola crenulata*	Root	Water	Fly	3–5 d	Wild-type *w^1118^*	Fed with food containing pathogenic microorganism (*S. marcescens*, *M. luteus*, and spores of *B. bassiana*) and toxic compound (SDS)	2.5% medium	Increased survival rate Increased expression of AMPs genes (*Defensin*, *d* and *Diptericin*) Decreased reactive oxygen species and epithelial cell death resulting the improvement of intestinal morphology	[56]
Phytohemagglutinin from *P. vulgaris*	-	-	Fly	3–7 d	Wild-type *w^1118^*	Fed with Phytohemagglutinin followed by systemic *Drosophila* C virus (DCV) infection	60–120 μg/mL	Increased survival rate Increased mRNAs of phagocytosis receptors Increased expression of AMPs genes (*Defensin* and *Diptericin*)	[57]
*Punica granatum*	Fruit (Juice)	-	Fly	2–3 d	Wild-type Canton-S strain	Pricked *C. albicans* suspension in the thoracic region and fed with food containing *Punica granatum* juice	0.1–15% medium	Increased survival rate Increased reproduction rate	[71]
*Hibiscus sabdariffa* L.	Calyx	Ethanol, water, ethyl acetate	Fly	4–7 d	*Psh^1^;;modSP^KO^* double mutant	Pricked *S. aureus* in the thorax	0.5–8% medium	Increased survival rate Increased the inhibition of bacterial proliferation	[72]
*Zingiber officinale*	Stem	-	Fly	5 and 33 d	Oregon strain	Fly was subjected to 30% hydrogen peroxide in food, fed with food containing DTT and ginger	25 mg/g medium	Diminished negative effects of DTT Increased life span of flies Increased locomotor performance (negative geotaxis)	[63]
*Garcinia kola*	Seed	Ethanol	Fly	1–3 d	Wild-type Harwich strain	Fed with food containing both Kolaviron and rotenone	100–500 μg/mL medium	Increased lifespan Improved AChE activity Increased Negative geotaxis Increased total thiols and GST level	[73]
*Vitex trifolia* L.	Leaf	Ethanol	Fly	4–7 d	*S. aureus*-infected flies and Wild-type *w^1118^*	Fed with food containing ethanol extract of legundi leaves	1–25% medium	Increased Survival rate Decreased bacterial propagation	[74]
*Rosmarinus officinalis* L.	Not mentioned	-	Fly (Male)	2 d	Oregon-R-C strain	Fed with high-fat diets containing rosemary extract	0.2–1.5 mg/mL medium	Increased Maximum lifespan Partially improved locomotor performance Increased antioxidant enzyme activity	[75]
*Centella asiatica* L.	Leaf	Acetone	Fly	Not mentioned	*elav-GAL4* strain and *UAS-Hsap/SNCA.F* strain	Fed with food containing *C. asiatica* extract	0.2–1 mg/mL medium	Decreased oxidative stress in the brains Delayed the loss of locomotor performance	[76]
*Bacopa monnieri*	Leaf	Acetone	Fly	Not mentioned	*elav-GAL4* strain and *UAS-Hsap/SNCA.F* strain	Fed with food containing *B. monnieri* extract	0.2–1 mg/mL medium	Decreased oxidative stress and apoptosis in the brains Delayed the loss of locomotor performance Decreased LPO level	[77]
*Vitis vinifera* (Grape)	Skin	-	Fly	Not mentioned	*Mhc*-*GAL4* strain, *dPINK1 RNAi*; *UAS*-*S6K OE* strain, *rictor*Δ*2*/*Y*; *Mhc*-*GAL4* strain, and *dPINK1 RNAi* strain	Fed with food containing grape skin extract powder or resveratrol	4–16% medium	Increased lifespan Rescued abnormal wing posture Removed mitochondrial aggregates	[68]
*Aronia melanocarpa*	Not mentioned	Acetone	Fly	1–3 d	Canton-S strain	Fed with food containing *A. melanocarpa* extract	1 and 2.5 μg/mL medium	Increased lifespan Ameliorated locomotor activity Decreased ROS and LPO level	[78]
*Rubus chamaemorus* L.	Fruit	Acetone	Fly	30 d	Canton-S strain	Fed with food that was applied with yeast paste containing *R. chamaemorus* extract at the top	0.12–0.6 mg/mL of medium	Increased lifespan Decreased aging rate Geroprotective effect was found in female flies Effect on male flies was weak	[79]
*Chrysantemum indicum* L.	Capitulum	Deionized water	Larva, Pupa, Fly	5 d	*Da-GAL4* strain, *S6k^l−1^/TM6B* strain and Wild-type *w^1118^* strain	Fed with high-sugar diet containing *C. indicum* extract	5–10% of medium	Increased lifespan Increased body weight and pupal volume Increased larval development Increased female flies fertility Decreased lipid accumulation	[80]
*Ipomoea batatas* L.	Not mentioned	-	Fly (Male)	3 d	Oregon-K strain, *esg-GAL4* strain and *UAS-GFP* strain	Fed with food containing *I. batatas* extract	0.5–2 mg/mL medium	Improved gut tissue homeostasis and prolonged lifespan Increased locomotor performance and oxidative stress tolerance Decreased aging rate Increased antioxidant enzyme activity and gene expression	[81]
*Vigna angularis*	Bean	Ethanol	Fly		*elav-GAL4* strain and *UAS* transgene *human* Aβ_42_ strain		1 mg/mL medium	Increased lifespan Increased locomotor performance Inhibited Aβ_42_ aggregates formation Suppressed cognitive impairment	[82]
*Bougainvillea glabra*	Leaf	Ethanol	Fly (Male)	1–4 d	Wild-type	Fed with food containing paraquat (neurotoxic) and *B. glabra* extract	40–200 μg/mL medium	Increased survival rate and locomotor performance Inhibited dopamine level reduction	[83]
*Gardenia jasminoides*	Not mentioned	-	Fly (Male)	3–10 d	*elav-GAL4^c155^* strain and *UAS-hAβ_42_* strain	Fed with food containing *G. jasminoides* extract for 5 h and resumed to normal food	5–500 g/mL medium	Rescued memory deficit Improved memory function	[84]

Abbreviations: AMPs: Antimicrobial Peptides; SDS, sodium dodecyl sulfate; ROS: Reactive Oxygen Species; JNK/SAPK: Jun amino-terminal kinases/stress-activated protein kinases; AChE: Acetylcholinesterase; NO: Nitric oxide; DSS, dextran sulfate sodium; PGRP-LB: Peptidoglycan recognition protein LB; NF-κB: Nuclear factor kappa B; DTT: Dithiothreitol; GST: Glutathione S-transferase; LPO: Lipid hydroperoxide.

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
