# Peer review of "Drosophila as an Animal Model for Testing Plant-Based Immunomodulators"

_ijms, 2022, doi:10.3390/ijms232314801_

Round 1
Reviewer 1 Report
The selected topics quite interesting to be connected with the urgency of testing natural products efficacy, especially on human related immune disorder. The review was quite fair enough with the good number of related research. However, the connection between the study using the Drosophila that continued to the next in vivo model was not yet well elaborated. How it can help/ support the next animal-based experiment design, how it provide quite reliable results/ contradictory, etc. Several minor typo should be easily eliminated. Please find the other detailed comments on the attached file. Good luck!

Author Response
Thank you very much for your precious comments. I would be grateful if you could find the attached file that contains our point-to-point answers.

Reviewer 2 Report
The manuscript is good and certainly contributes to scientific development. Drosophila is already been explored as an infectious and neurological model, so some light on this aspect can also be included .
the manuscript itself is Ok but while sutdyng immunomodulatory role , we also need to study the effect of the extract in prophylaxis and treatment of infections as well. so. if the authors can include those studies also in their review, the readers get a complete information on the use of drosophila in immunological studies. apart from this, drosophila is also used for neurological studies which can also be a part of immunomodulatory signalling pathway. presently, most of the researchers are using either laboratory animals or in some cases experimental animals, which require manpower and prove to be cost and time consuming. drosophila cna certainly be a goood alternative for them.So if these aspects are also included, the scope of readers and viewers can be significantly increased.
Author Response

(The authors gave the same response as above.)

Round 2
Reviewer 1 Report
Thank you for the confirmation about the raised issue. There are no other question about it. Good luck for the further research.